# Enhanced Message Replication Technique for DTN Routing Protocols

**DOI:** 10.3390/s23020922

**Published:** 2023-01-13

**Authors:** Siham Hasan, Meisam Sharifi Sani, Saeid Iranmanesh, Ali H. Al-Bayatti, Sarmadullah Khan, Raad Raad

**Affiliations:** 1Faculty of Technology, De Montfort University, Gateway House, Leicester LE1 9BH, UK; 2Sydney International School of Technology and Commerce, Sydney, NSW 2000, Australia; 3School of Electrical, Computer and Telecommunication Engineering, University of Wollongong, Wollongong, NSW 2522, Australia

**Keywords:** delay-tolerant network, network congestion, DTN routing, congestion control, wireless networks, ad hoc networks

## Abstract

Delay-tolerant networks (DTNs) are networks where there is no immediate connection between the source and the destination. Instead, nodes in these networks use a store–carry–forward method to route traffic. However, approaches that rely on flooding the network with unlimited copies of messages may not be effective if network resources are limited. On the other hand, quota-based approaches are more resource-efficient but can have low delivery rates and high delivery delays. This paper introduces the Enhanced Message Replication Technique (EMRT), which dynamically adjusts the number of message replicas based on a node’s ability to quickly disseminate the message. This decision is based on factors such as current connections, encounter history, buffer size history, time-to-live values, and energy. The EMRT is applied to three different quota-based protocols: Spray and Wait, Encounter-Based Routing (EBR), and the Destination-Based Routing Protocol (DBRP). The simulation results show that applying the EMRT to these protocols improves the delivery ratio, overhead ratio, and latency average. For example, when combined with Spray and Wait, EBR, and DBRP, the delivery probability is improved by 13%, 8%, and 10%, respectively, while the latency average is reduced by 51%, 14%, and 13%, respectively.

## 1. Introduction

Delay-tolerant networks (DTNs) are intermittently connected (wireless) networks, wherein there is no guaranteed end-to-end connectivity between the source and destination nodes [1,2,3,4]. These networks can be disrupted by factors such as node mobility, density, and limited radio range, leading to frequent disconnections. As a result, traditional Transmission Control Protocols/Internet Protocols (TCPs/IPs) cannot be used, and instead, a store–carry–forward approach is employed to allow data transmission to continue despite the lack of a continuous path [4,5]. However, this routing approach can cause issues such as buffer congestion and inefficient use of network resources due to the dissemination of multiple copies of a message across the network. [4,6,7].

There are several routing techniques [8,9,10,11,12,13,14,15,16,17,18] available for addressing the routing problem in delay-tolerant networks (DTNs), which can be divided into two categories: flooding-based and quota-based [19,20]. Flooding-based protocols allow nodes to replicate messages indefinitely, which can lead to good network performance when resources are not limited. However, they may have a low delivery ratio, high overhead, and high delay when resources are limited [21,22]. Quota-based protocols, on the other hand, set a maximum number of replicas for each message—unlike flooding-based protocols, where the number of copies is determined by the number of encounters [20]. These protocols are resource-efficient and perform well when resources are limited but may experience low delivery rates and long delays when resources are insufficient. Additionally, the number of replicas is fixed for all messages and is not affected by network capacity, such as node buffer size, energy, time-to-live (TTL) values, and encounter rate [21,22].

The limitations of both flooding-based and quota-based routing protocols in DTNs have led us to propose the Enhanced Message Replication Technique (EMRT). The EMRT dynamically adjusts the number of message replicas based on a node’s ability to quickly disseminate the message, taking into consideration various parameters, such as existing connections, encounter history, buffer size history, energy, and TTL values. This can improve the delivery ratio and enhance the network resource utilization. The main contributions of this paper are as follows:

This is the first study to focus on optimizing the parameters of quota-based routing protocols in delay-tolerant networks (DTNs). Our focus is on the initial number of replicas, which can be increased or decreased. It is worth noting that existing quota-based protocols assume a fixed maximum initial number of replicas for each generated message.We propose a heuristic called the Enhanced Message Replication Technique (EMRT) to improve the performance of existing quota-based routing protocols in delay-tolerant networks (DTNs) by adaptively adjusting the maximum initial number of replicas for each generated message based on network circumstances such as density, buffer availability, energy, and TTL values.Simulation results show that applying the EMRT over well-known quota-based routing protocols improves the network performance in terms of delivery ratio, overhead, and latency average compared to the original forms of existing protocols.

The remainder of this paper is organized as follows: Section 2 describes the related works. Section 3 proposes the Enhanced Message Replication Technique (EMRT). The simulation studies and their results are presented in Section 4. Finally, Section 5 concludes the paper.

## 2. Related Works

### 2.1. Flooding Protocols

A common DTN flooding protocol is the Epidemic routing protocol that was proposed by Vahdat and Becker [23]. The Epidemic protocol transmits the messages between the nodes in the network without any restriction. Due to the limited DTN network resources and unlimited message replicas disseminated through the network, the Epidemic routing protocol has proven less sufficient in the congested traffic network, leading to dropped messages and high overhead. To address the issues associated with the Epidemic routing protocol, Matsuda and Takine [24] proposed the (*p*–*q*) Epidemic algorithm with a vaccination routing protocol (PQERPV). The *q* probability indicates the probability of receiving a message from the source, while the *p* probability indicates the probability of receiving a message from other nodes. The performance of the PQERPV protocol is influenced by the value of *q* and the *p*-values, which can impact the protocol’s performance. Lower *p*-values can lead to a low rate of message dissemination. In [25], the two-hop forwarding protocol was proposed by Grossglauser. In this routing protocol, the sender nodes exchange bundles with arbitrarily encountered nodes that will forward these messages only to the destination node. The overhead of this protocol is lower than the overhead of pure Epidemic since the message is sent across two hops only. However, if the destination node is not reachable via two hops, the messages may fail to be delivered to the destination node. Furthermore, due to the low dissemination rate, the message may suffer from a large delay. Additionally, various DTN protocols that select the next hop node based on encounter history have been proposed to decrease the network overhead issue of flooding protocols. One example of these protocols is PROPHET, which was presented by Lindgren et al. [13]. PROPHET utilizes the delivery predictability parameter to estimate the probability of delivering the messages to their destination. In PROPHET, the delivery predictability parameter is updated after each node’s encounter. However, when a node encounters a neighboring node with low delivery predictability, there is no guarantee that it will encounter another node with a higher delivery predictability value during the message lifetime; therefore, the messages may never leave the source node. On the other hand, if the source encounters many nodes with high delivery probability, the overhead will be high because the messages are flooded through the network. MaxProp is another example of a flooding-based routing protocol, which achieves better performance in terms of high delivery rates and low average latency [9]. However, MaxProp may produce a high overhead on restricted-resource networks [6]. Moreover, in the RAPID protocol [8], the messages are ordered using a utility function. Each message is assigned a utility by the RAPID protocol. RAPID replicates messages that lead to an increase in utility [8]. One major drawback of RAPID is that replica information must be flooded throughout the network to derive the utility of messages, leading to high overhead and high delivery delays. Additionally, the information disseminated may be outdated when it reaches the nodes because of delays. PREP is a prioritized Epidemic routing protocol where the messages are assigned with priority based on both the cost to the destination and the expiration time. This assigned priority is used to determine which messages should be sent or dropped when the buffer or bandwidth is limited. Erramilli and Crovella [10] proposed techniques to utilize the properties of nodes to make forwarding decisions by selecting the optimal node to forward messages based on utility value. However, this technique causes flooding when many encountered nodes have high utility values. Moreover, the messages may never leave the source nodes when many of the encountered nodes have low utility values.

Based on the above analysis, the issue with flooding-based protocols is the high consumption of network resources such as bandwidth, buffer, and energy. Therefore, to mitigate the waste in the network resources issue, quota-based protocols should be investigated.

### 2.2. Quota-Based Protocols

One example of a quota-based protocol is the single-copy scheme proposed by Spyropoulos et al. [26], in which the source delivers messages directly to the destination node. However, this protocol can be redundant if the destination is located far from the source node, and the messages may never be delivered. To address this issue, Spyropoulos et al. [15] proposed the Spray and Wait routing protocol, which is a multi-copy two-hop scheme. In the Spray and Wait protocol, the number of replicas is fixed at the time of message creation. The protocol consists of two phases: the spray phase, in which message replicas are disseminated, and the waiting phase, in which the node waits with a single replica message to meet the destination. While the Spray and Wait protocol is resource-efficient, it may still fail to deliver messages if the destination is in a different area than the source node. To address this issue, Spyropoulos proposed the binary Spray and Wait routing protocol, in which the node forwards half of the message’s replicas during each contact. Cui et al. [27] proposed the Adaptive Spray and Wait (QoN-ASW) routing algorithm, which adaptively allocates the number of message copies between encountered nodes based on the quality of node (QoN) metric during the spray phase. In the waiting phase, a forwarding scheme is implemented to increase flexibility. Unlike the direct transmission approach, the QoN-ASW strategy makes use of encounter opportunities and, when a node is left with only one copy, it forwards the copy to a suitable candidate node with higher delivery predictability instead of waiting for the destination to be encountered. While the QoN-ASW algorithm improves network performance in terms of delivery rate and average delay, it has a slightly higher overhead than the Spray and Wait protocol.

Another example of a quota-based protocol is Spray and Focus, which was proposed by Spyropoulos et al. [16]. This protocol uses a timer-based utility value to track the intervals between nodes’ encounters and assumes that nodes with similar mobility patterns have short intervals between encounters. Spray and Focus also has two phases: the spray phase, and the focus phase, in which a single copy is forwarded to maximize a utility function. Both Spray and Wait and Spray and Focus aim to reduce the high overhead caused by flooding-based protocols but have low delivery ratios due to low message dissemination [28]. Bulut et al. [29] proposed an algorithm for transmitting replicas over various periods, in which the source node forwards *n* copies to the first *n* encountered nodes and waits for an acknowledgment to confirm successful delivery. If delivery fails, more copies are injected into the network in subsequent periods to increase the delivery probability. However, if a message is successfully delivered, the source node’s acknowledgment may not be received on time due to the significant delays in the DTN, leading to the forwarding of many replicas to nodes and causing buffer congestion and dropped messages. Dhurandher et al. [30] proposed the Encounter- and Distance-based Routing (EDR) protocol for DTNs, which uses a forward parameter to select the next hop based on the number of encounters and distance to the destination. However, EDR assumes that all nodes have sufficient energy, which may not be the case, and it was not evaluated in terms of delivery ratio—a crucial evaluation metric in DTNs. In [31], the Adaptive Message Replication Technique (AMRT) was proposed for use with quota-based protocols. The AMRT controls network traffic by assigning different numbers of replicas for each generated message based on network conditions (such as congestion among the sender’s neighbors), using historical traffic information as an estimate of future capacity. The AMRT was implemented and evaluated with various quota-based routing protocols, improving the delivery ratio and reducing the delivery delay.

Another work in the field of quota-based routing protocols is Encounter-Based Routing (EBR), proposed by Nelson et al. [21]. EBR creates a limited number of replicas for each message based on the history of a node’s encounters. In EBR, nodes that have encountered one another frequently have the best chance of successfully transmitting messages to the destination. However, a destination may never receive transmitted messages if it has low encounter rates in a low-density network. To address this issue, Iranmanesh et al. [22] proposed the Destination-Based Routing Protocol (DBRP). DBRP assigns weights to nodes based on the rate of encounters with the destination and other nodes, with nodes that have encountered the destination receiving higher weights. In networks with limited node buffer space, the DBRP has been shown to perform better than other quota-based routing protocols, such as Spray and Wait and EBR. However, the DBRP does not consider a node’s capability in determining the appropriate number of replicas, which can lead to either buffer congestion due to lack of space or inefficient use of network resources when a small number of replicas are transmitted by a node with sufficient resources. Both EBR and the DBRP have been shown to have good performance through simulation, but blindly forwarding replicas to nodes with high encounter rates without considering their capability can lead to delivery delays, message drops, and inefficient resource utilization. In [32], a novel method was proposed to address the issue of resource consumption caused by message replication. This method determines the dissemination of DTN data based on the expected path of a node, transmitting duplicates to nodes that are close to the destination. The method was evaluated by comparing the number of arrived data to the number of generated data for both the proposed method and an existing method, and the former was found to achieve a higher message arrival rate. However, this method focuses on where duplicated messages should be sent but does not consider how many replicas of each message should be sent to encountered nodes.

Although quota-based protocols are more resource-friendly than flooding-based protocols, they still suffer from low delivery ratios and high delivery delays. To address this issue, we propose the Enhanced Message Replication Technique (EMRT) protocol—a dynamic quota-based technique that considers not only encounter-based routing metrics, but also network congestion and capacity, to minimize overhead, maximize the delivery ratio, and efficiently utilize network resources. Table 1 provides a summary of the features and characteristics of DTN routing protocols [33]. The table includes details on both flooding-based and quota-based DTN routing protocols.

## 3. Enhanced Message Replication Technique (EMRT)

The main objective of DTN routing protocols is to achieve high delivery ratios, low delivery delay, and low overhead [30]. Based on the literature review, quota-based protocols suffer from trial-and-error values for the parameters of the algorithm. In this section, we propose the EMRT technique to set the maximum initial number of replicas for each generated message dynamic, based on network circumstances such as density, buffer availability, energy, and TTL values.

To define each of the criteria above, we assume that each node has an encounter value (EV), which indicates the history of the rate of encounters. This implies that the past rate of encounters can be used to predict the future encounter rate. Let us briefly explain how EVs are calculated: Every node should maintain a current window counter (CWC) that counts the number of encounters in the last time interval. Let us assume that each time interval duration is 30 s. Accordingly, the EV of a source node *s* is updated as follows:(1)$$EV_{s}=\alpha \times CWC+\left(1-\alpha \right)\times EV_{s}$$
where α ∈ (0,1), i.e., 0.85. When the interval has ended, the CWC is reset to zero, and the EV is updated. In order to evaluate whether an encountered node is likely to be located in a high-density area, we can use the criterion of EV. Based on this information, the initial number of replicas for a new message can be appropriately determined for the node. In addition to the encounter rate, we have also defined other criteria to consider when determining the initial number of replicas.

The average buffer of nodes that the source node has encountered is another criterion, which is represented by $B_{avg}$. We need to consider this information to evaluate the space capacity of the nodes that the source will likely see again in the future. If their storage has enough space, more replicas will be created. Otherwise, fewer initial replicas will be generated to reduce the rate of dropped messages.

Time-to-live, or $TTL_{i}$, is another criterion that represents the amount of time that a generated message *i* can remain in the network. If the TTL expires, the message will be dropped. Hence, if the TTL is short, the dissemination should be more. Consequently, the maximum initial number of replicas should be increased.

Lastly, $E_{s}$ represents the available energy of the source node. If the remaining energy is small and the node is about to die, a message generated by that node cannot live for a long time at the source node. Thus, it should be disseminated to other encountered nodes. As a result, if $E_{s}$ is small, the maximum initial number of replicas should be increased.

Based on the above definitions, after normalization, the maximum initial number of replicas for each generated message *i* can be calculated as follows:(2)$$M_{i}=m_{init}\times \frac{EV_{s}+B_{avg}}{TTL_{i}+E_{s}}$$
where $m_{init}$ is the initial number of message replicas that the existing quota-based routing protocols use. As a result, $M_{i}$ will be the new maximum initial number of replicas for the quota-based protocol.

Algorithm 1 shows how each source node produces a different number of initial replicas for each generated message. As mentioned above, each node is responsible for maintaining the encounter value, which is used to predict future encounter rates (Line 2). The CWC is used to count the number of encounters in the current time interval. Thus, when the update interval expires, the CWC will be reset to zero (Line 3). For each generated message, the maximum initial number of replicas is calculated based on Equation (2) (Line 7). This is while $m_{init}$ is dependent on the routing protocol used. For example, EBR uses $m_{init}=11$, while Spray and Wait uses $m_{init}=8$. Consider the numerical example in Figure 1. Let us assume that the routing protocol used is EBR. For node A, $EV_{s}=20$, $B_{avg}=25$, *TTL_i_* = 8, $E_{s}=11$, and $m_{init}$=11. $M_{i}$ for node A is $M_{i}=11*\;\frac{20+25}{8+11}=22$. This is while, based on the criteria of node B, $M_{i}=11*\;\frac{12+20}{20+30}=4$.

**Algorithm 1**: EMRT1:***if****time* ≥ *nextUpdate*
***then***

2: EV ← α∗ CWC+(1 − α)∗ EV



3: CWC ← 0



4: nextUpdate ← time+Wi



5: end if



6: for every new messages i do



$7:\;M_{i}\;\leftarrow \;m_{init}\times \frac{EV_{s}+B_{i}}{TTL_{M}+E_{s}}$



8: end for



## 4. Simulation Studies

The EMRT has been tested using the Opportunistic Network Environment (ONE) simulator, which is a Java-based open-source tool for evaluating and implementing routing protocols, particularly for DTNs [34]. The performance of the EMRT was evaluated against the original forms of Spray and Wait, EBR, and the DBRP. The experiments were conducted based on the benchmark simulation settings available in the ONE simulator, as listed in Table 2.

### 4.1. Metrics’ Performance

The goal of the experiments was to test the impact of the EMRT on the Spray and Wait, EBR, and DBRP quota-based protocols by comparing the results with the original versions of these protocols. This means that the performances of these protocols were not compared with one another, but rather, the performance of the protocols with and without the EMRT was compared in order to validate the effectiveness of the EMRT. Six metrics, as defined in [31], were used in the analysis.

The delivery ratio is a key metric used to evaluate the performance of a network. It is calculated as the ratio of the number of delivered messages (NDB) to the number of generated messages (NGB), as shown in Equation (3):(3)$$\mathrm{Delivery}\;\mathrm{ratio}=\frac{NDB}{NGB}$$

The overhead is defined as the ratio of the number of delivered messages (NDB) to the number of relay nodes (NRN), as shown in Equation (4):(4)$$\mathrm{Overhead}\;\mathrm{ratio}=\frac{NDB-NRN}{NDB}$$

The latency average, measured in seconds, is the average time it takes for messages to be delivered to their destination in the network. Unlike other network parameters such as the delivery ratio and overhead, which are expressed as ratios and do not have units, the latency average is given by Equation (5):(5)$$\mathrm{Latency}\;\mathrm{average}=\frac{\sum_{i=1}^{NDB}t_{i}}{NDB}$$

In order to account for the trade-off between different metrics and penalize protocols that unfairly optimize a single metric, three composite metrics—DL, DO, and DLO—were used to compare the delivery probability and conventional metrics as ratios. These metrics, which were introduced in [30,31], are defined as follows:

DL is a composite metric that is used to adjust the performance of a protocol if it is optimized for delivery ratio but has poor latency. DL is a ratio of the delivery ratio (DR) to the latency average (LA), and it can be calculated using Equation (6):(6)$$\mathrm{DL}=\mathrm{DR}\;\times \;\frac{1}{\mathrm{LA}}$$

As defined in Equation (7), DO combines the DR and the overhead ratio (OR) to capture the trade-off between these two metrics.
(7)$$\mathrm{DO}=\mathrm{DR}\;\times \;\frac{1}{\mathrm{OR}}$$

Lastly, the trade-off between DR, LA, and OR is captured by DLO, as shown in Equation (8):(8)$$\mathrm{DLO}=\mathrm{DR}\;\times \;\frac{1}{\mathrm{LA}}\;\times \;\frac{1}{\mathrm{OR}}$$

### 4.2. Simulation Results

The performance of the EMRT was analyzed by varying the node density (i.e., number of nodes). The results of the impact of the EMRT on Spray and Wait, EBR, and the DBRP at various node densities are shown in Figure 2, Figure 3 and Figure 4. To clarify, “Spray & Wait-EMRT”, “EBR-EMRT”, and “DBRP-EMRT” refer to the application of the proposed EMRT on the original Spray and Wait, EBR, and DBRP quota-based routing protocols, respectively. The performance of Spray and Wait–EMRT, EBR–EMRT, and DBRP–EMRT is improved by the EMRT when the number of nodes exceeds 50. This is because the source node adaptively adjusts the initial maximum number of message replicas according to its own buffer size, allowing it to generate the appropriate number of copies for each message rather than relying on a fixed, limited number of copies for all generated messages.

Figure 2 shows the relationship between the node density and message delivery ratio for different quota-based protocols. The delivery ratio for all protocols (i.e., the original Spray and Wait, EBR, and DBRP, as well as the enhanced Spray and Wait–EMRT, EBR–EMRT, and DBRP–EMRT versions) increases significantly as the number of nodes increases. This is because when there are more nodes, more of them are involved in delivering messages, and the number of replicas also increases, increasing the chance of the messages being delivered. In addition, Figure 2 shows a significant improvement when the proposed EMRT technique is applied to Spray and Wait, with results showing an improvement of up to 13% compared to the original Spray and Wait. This is because Spray and Wait relies on a fixed number of replicas for all messages, while Spray and Wait–EMRT uses a dynamic number of replicas based on the node’s ability to carry the messages without causing congestion or wasting network resources. As a result, the EMRT reduces the number of copies of newly generated messages when the source’s neighbors experience message congestion. The EBR–EMRT protocol also shows improvement when the EMRT is applied, with the delivery ratio increasing by up to 8% compared to the original EBR. This is because, in addition to directing replicas towards high-density areas, EBR–EMRT controls the number of replicas based on the capability of the node, taking advantage of the available size of the node buffer to carry and store more copies of the message until a good forwarding opportunity arises. Finally, the DBRP–EMRT also achieves an improvement of up to 10% compared to the original DBRP by controlling the level of congestion in the network to improve the delivery ratio, by ensuring that the number of replicas is appropriate for the available network resources.

Figure 3 shows the relationship between the overhead ratio and node density. As the node density increases, the overhead ratio decreases for all protocols, because these quota-based protocols rely on a small number of message copies, so they are not greatly affected by the increasing number of nodes. At the same time, the figure also shows the results of applying the EMRT to Spray and Wait, EBR, and the DBRP in terms of the overhead ratio. There is no significant reduction in the overhead ratio of DBRP–EMRT, while the overhead ratios of the Spray and Wait–EMRT and EBR–EMRT protocols are reduced by up to 9%, and 6%, respectively, compared to Spray and Wait and EBR. This is because the network resources are controlled and congestion is reduced by relaying messages based on the rate of encounter and the level of network congestion (i.e., available network resources), resulting in a low overhead ratio. This means that when a node has sufficient resources, such as free buffer space and energy, the number of replicas for newly generated messages will be increased. Otherwise, the number of copies of newly generated messages will be decreased.

Figure 4 shows the relationship between the latency average and node density, where it is clear that the latency average increases as the number of nodes increases. This is because more nodes become involved, leading to congestion in the network, and information processing at each node takes time, resulting in an overall increase in latency. In addition, buffering messages at a node causes delays in message delivery. The figure also shows the results of applying the EMRT to Spray and Wait, EBR, and the DBRP in terms of latency average. The increased latency average at higher densities with the original protocols is due to more nodes sending copies of the message, leading to a delay in message delivery. It can be seen that the latency average of the Spray and Wait–EMRT, EBR–EMRT, and DBRP–EMRT protocols is reduced by up to 51%, 14%, and 13%, respectively. This is because the rate of dropping messages is controlled by selecting a number of replicas appropriate for the available network resources. An uncontrolled number of replicas can lead to buffer congestion or inefficient use of network resources. By reducing congestion, the rate of message drops is reduced, resulting in a decrease in the latency average.

Figure 5 shows the composite metric combining delivery ratio and latency average. The results show that applying the EMRT to Spray and Wait can improve both the delivery ratio and the latency average by up to 60%. This is due to the individual improvements in each of these metrics, as explained in Figure 2 and Figure 4, respectively. Additionally, there is an improvement of up to 25% in this composite metric for EBR and the DBRP. These improvements are less compared to Spray and Wait, and this is due to the more targeted forwarding strategy used in these protocols.

The composite metric in Figure 6, which evaluates both delivery ratio and overhead, shows an improvement of up to 14% for all considered protocols when the EMRT is applied. The EMRT increases the number of replicas when there are sufficient resources, leading to an expected increase in overhead. However, when there are insufficient resources, the number of replicas decreases. Therefore, the EMRT is not expected to lead to a significant improvement in the overhead ratio. However, the high dissemination rate resulting from the production of more replicas increases the delivery ratio, leading to the impact shown in Figure 6. This trade-off between overhead and delivery ratio is why the EMRT has an impact on this composite metric.

Finally, Figure 7 shows a clear image of the performance of the EMRT when applied to the considered protocols and the delivery ratio, average latency, and overhead are all combined. The results for all protocols show an improvement of up to 72%. This is because the improvement of applying the EMRT can be seen in all individual metrics as well. The EMRT results show that making the number of replicas adaptive increases the overall performance of the routing protocol. This is because the number of copies is not dependent on the number of encounters and remains fixed for each generated message.

## 5. Conclusions

This paper introduces the Enhanced Message Replication Technique (EMRT)—a novel message replication technique for quota-based protocols that enables the number of replicas to be dynamic based on the nodes’ capabilities. The EMRT is designed to generate the optimal number of replicas for newly generated messages by considering the history of encounters, as well as the buffer availability, energy, and time-to-live values of nodes. As a result, the EMRT improves the performance of quota-based protocols and more effectively utilizes network resources. To evaluate the performance of the EMRT, three quota-based protocols (Spray and Wait, Encounter-Based Routing, and the Destination-Based Routing Protocol) were considered. The results show that the EMRT significantly improves the delivery ratio (up to 13%, 8%, and 10% for Spray and Wait, EBR, and the DBRP, respectively) and reduces the latency average (by 51%, 14%, and 13% for Spray and Wait, EBR, and the DBRP, respectively) compared to the original quota-based protocols.

## Figures and Tables

**Figure 1 sensors-23-00922-f001:**
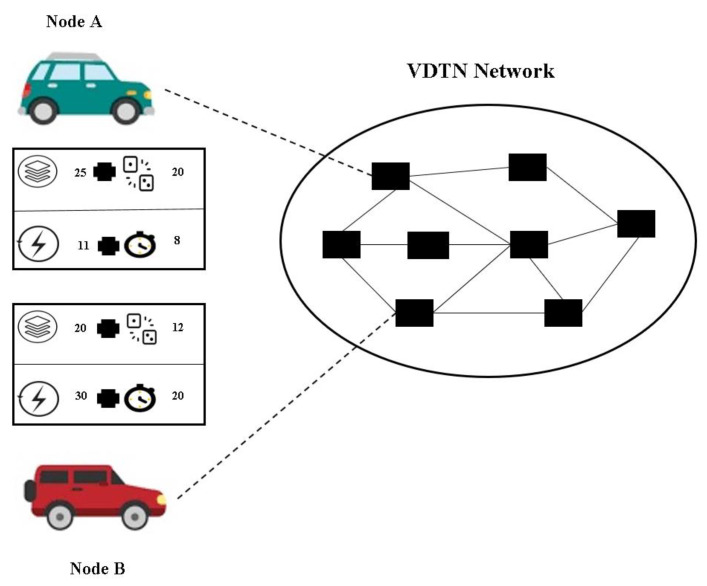
The network’s circumstances from each node’s perspective.

**Figure 2 sensors-23-00922-f002:**
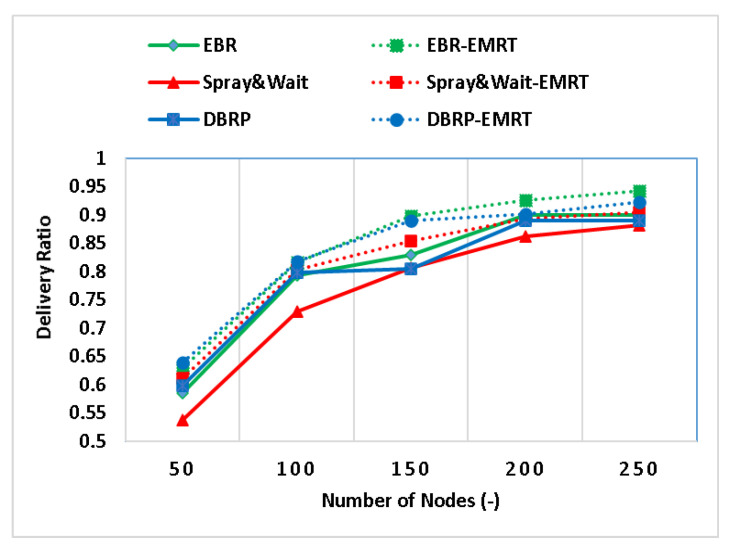
The delivery ratio with different node densities.

**Figure 3 sensors-23-00922-f003:**
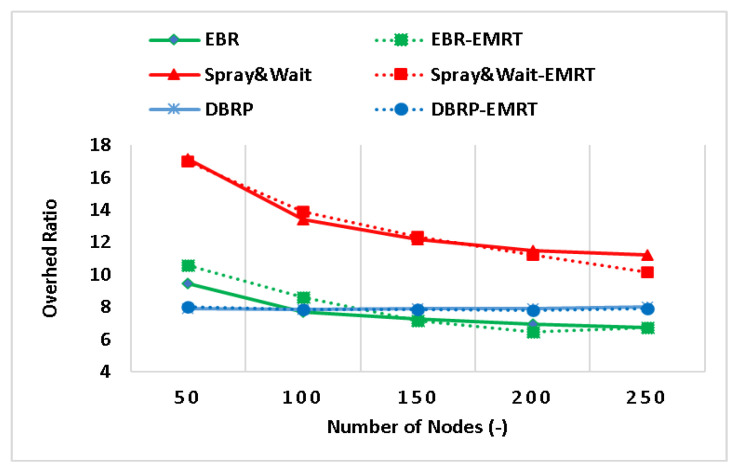
The overhead ratio with different node densities.

**Figure 4 sensors-23-00922-f004:**
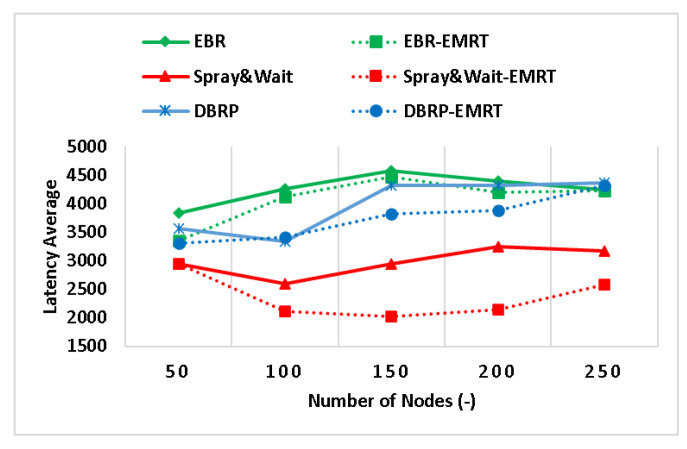
The latency average with different node densities.

**Figure 5 sensors-23-00922-f005:**
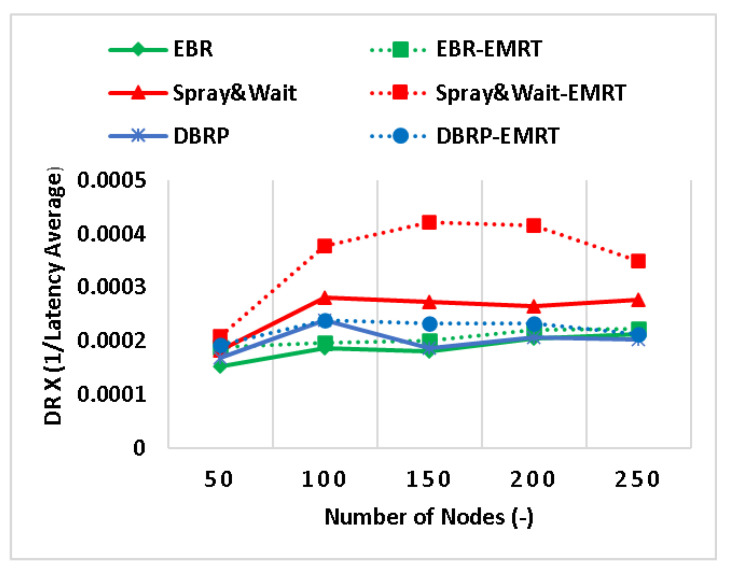
The DL composite metric (DR × (1/latency average)).

**Figure 6 sensors-23-00922-f006:**
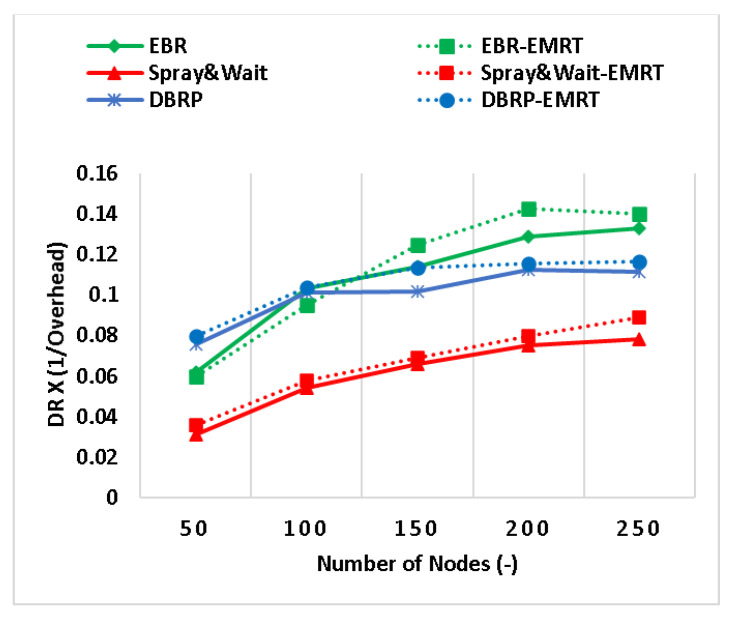
The DO composite metric (DR × (1/overhead)).

**Figure 7 sensors-23-00922-f007:**
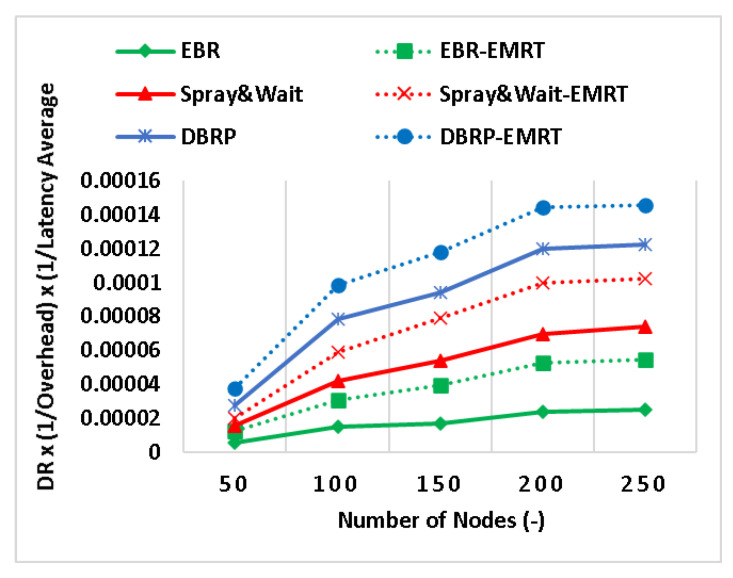
The DLO composite metric (DR × (1/overhead) × (1/latency average).

**Table 1 sensors-23-00922-t001:** Comparison of the various DTN strategies.

Protocols	Category	Decision Criteria	Advantages	Limitations	Delivery Ratio	Average Delay	Overhead
Epidemic [23]	Flooding	None	Simple; no prior knowledge required	High drop ratio High overhead ratio	High if resources are unlimited	Low if resources are unlimited	High
(*p*,*q*)-Epidemic [24]	Flooding	None	Recovery process to clear unnecessary messages No prior knowledge is required	High drop ratio with limited resources High power consumption	High if resources are unlimited	Low if resources are unlimited	High
PROPHET [13]	Flooding	History	Universal and based on the delivery probability	High drop ratio Acts like Epidemic Low delivery probability	High if resources are unlimited	Low if resources are unlimited	High
MaxProp [9]	Flooding	History	Less message traffic	High drop ratio with limited resources High power consumption	High if resources are unlimited	Low if resources are unlimited	High
Spray and Wait [15]	Quota	None	Simple and resource-friendly	High drop ratio if resources are limited High power consumption	High if resources are unlimited	Low if resources are unlimited	Medium
QoN-ASW [27]	Quota	History	Resource-friendly	High drop ratio if resources are limited High power consumption	High if resources are unlimited	Low if resources are unlimited	High
Spray and Focus [16]	Quota	History	Simple and resource-friendly	High drop ratio if resources are limited High power consumption	High if resources are unlimited	Low if resources are unlimited	Medium
Bulut et al. [29]	Quota	None	Simple; no prior knowledge required	High drop ratio if resources are limited High power consumption	High if resources are unlimited	Low if resources are unlimited	Medium
AMRT [31]	Quota	History	Resource-friendly	High drop ratio High power consumption	High if resources are unlimited	Low if resources are unlimited	Low
EBR [21]	Quota	History	Resource-friendly	High drop ratio High power consumption	High if resources are unlimited	Low if resources are unlimited	Low
DBRP [22]	Quota	History	Resource-friendly	High drop High power consumption	High if resources are unlimited	Low if resources are unlimited	Low

**Table 2 sensors-23-00922-t002:** Parameters of the simulation work.

Parameters	Value
Total Simulation Time	12 h
World Size	Helsinki, Finland, 5 × 3 km^2^
Movement Model	Map-based model
DTN Routing Protocol	Spray and Wait, EBR, DBRP
Speed of Nodes (m/s)	Tram: *U* (7,10)
	Vehicles: *U* (2.7: 13.9)
	Pedestrian: *U* (0.5: 1.5)
Buffer Size	20 MB
Number of Nodes	50, 100, 150, 200, 250
Date Rate	54 Mbps
Interface Transmit Range	140 m
Message Time to Live	60 min
Node Movement Speed	Min = 0.5 m/s Max = 1.5 m/s
Message Creation Rate	One message per 25–35 s
Message Size	100 KB

## Data Availability

Not applicable.

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
