# Peer review of "Enhanced Message Replication Technique for DTN Routing Protocols"

_sensors, 2023, doi:10.3390/s23020922_

Round 1
Reviewer 1 Report
The following revisions are required.
1. In literature review, add 3 to five more relevant and latest techniques.
2. Add Comparison table at the end of section 2 and compare with at least 10 to 15 techniques with appropriate parameters.
3. Please make sure your paper has necessary language proof-reading.
Reviewer 2 Report
From technical perspective, the paper represents solid contribution. However, before the acceptance, its structure should be revised and improved:
- Language improvements and re-writing unclear constructions, such as: “replicas dynamic based on the capability of node”.
- Avoid repetitive constructions, as “proposed EMRT was applied to quota protocol to improve the network performance”,…
- List of the contributions in the introduction could be shortened and summarized, maintaining only the most necessary elements
- Table 1 could be placed elsewhere, such as Related works or Background section
- Table 2 formatting is wrong
- Figures occupy too much space
Reviewer 3 Report
one or two figures on the scenarios would have made it more readable and meaningful.
Reviewer 4 Report
The paper consider the Delay Tolerant Networks (DTNs) and proposes an Enhanced Message Replication Technique (EMRT) to make the number of replicas dynamic based on the existing connections, history of encounters, history of buffer sizes, time-to-live values, and energy to disseminate message quickly.
The presented work makes some interesting points, but overall there are a lot of vital issues that question its merit.
To begin with, there are multiple typos, grammar and reference format mistakes throughout the paper, indicating that it was not carefully written. Here just highlight some of them, but the authors are advised to review their manuscript themselves to spot all of the mistakes. In the reference section, all of the reference papers do not show the associated journal name or conference name. In the paper, there are two section 4 (i.e. Evaluation section and Simulation Results section). In the page 8, The Equ. (4) is same to Equ. (5).
The paper proposes EMRT scheme. The key parameters, such as EV, encounter rate, Current Window Counter, Number of replicas of message, and the average buffer availability are not clearly definition. Thus it is difficult to understand the operations of the EMRT scheme. Furthermore, the paper based on the ONE simulator which is difficult to prof that the simulator is suit for this work or not.
Round 2
Reviewer 4 Report
The paper has revised according to the review's comments. It is fine to accept the paper in this version.